# Know Your Space: Inlier and Outlier Construction for Calibrating Medical OOD Detectors

**Vivek Narayanaswamy**[*1]                                          VNARAY29@ASU.EDU

**Yamen Mubarka**[*2]                                                MUBARKA1@LLNL.GOV

**Rushil Anirudh**[2]                                                ANIRUDH1@LLNL.GOV

**Deepta Rajan**[3]                                                  R.DEEPTA@GMAIL.COM

**Andreas Spanias**[1]                                               SPANIAS@ASU.EDU

**Jayaraman J. Thiagarajan**[2]                                      JJAYARAM@LLNL.GOV

[1] *Arizona State University, USA,* [2] *Lawrence Livermore National Labs, USA,* [3] *Microsoft, USA*

**Editors:** Accepted for publication at MIDL 2023

## Abstract

We focus on the problem of producing well-calibrated out-of-distribution (OOD) detectors, in order to enable safe deployment of medical image classifiers. Motivated by the difficulty of curating suitable calibration datasets, synthetic augmentations have become highly prevalent for inlier/outlier specification. While there have been rapid advances in data augmentation techniques, this paper makes a striking finding that the space in which the inliers and outliers are synthesized, in addition to the type of augmentation, plays a critical role in calibrating OOD detectors. Using the popular energy-based OOD detection framework, we find that the optimal protocol is to synthesize latent-space inliers along with diverse pixel-space outliers. Based on empirical studies with multiple medical imaging benchmarks, we demonstrate that our approach consistently leads to superior OOD detection (15% − 35% in AUROC) over the state-of-the-art in a variety of open-set recognition settings.

**Keywords:** Deep neural networks, out-of-distribution detection, data augmentation, energy, medical imaging, open-set recognition.

## 1. Introduction

Detecting out-of-distribution (OOD) data characterized by a variety of semantic or covariate shifts with respect to the in-distribution (ID) data is vital for safe adoption of AI tools in medical imaging (Hosny et al., 2018; Young et al., 2020). As a result, a broad class of inference-time, scoring functions (Hendrycks and Gimpel, 2017; Liang et al., 2018; Lee et al., 2018; Liu et al., 2020; Sastry and Oore, 2020; Ren et al., 2021) that can reliably distinguish between ID and OOD data has emerged. However, in practice, one needs to calibrate those detectors, such that the dual objective of sufficiently generalizing to ID test data and reliably rejecting OOD data is effectively met.

Existing approaches for calibrating OOD detectors require users to specify regimes of inlier and outlier data. For example, a popular approach for specifying inliers is to leverage synthetic augmentations (Shorten and Khoshgoftaar, 2019) that produce plausible ID data variations. Typical choices include geometric transforms such as rotation and

---

[*] Contributed equally

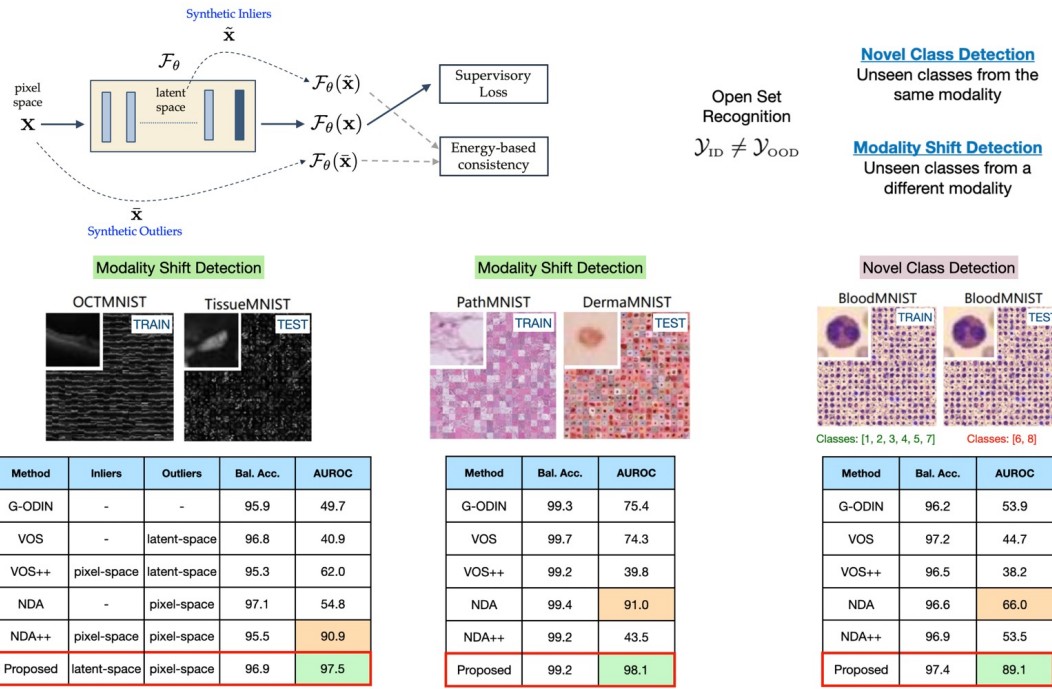

Figure 1: **Specifying synthetic inliers/outliers to calibrate OOD detectors**. We focus on energy-based OOD detectors in deep models and explore synthetic augmentations for specifying calibration data. We make a striking finding that the space in which the augmentations are synthesized plays a critical role on the detection performance. While state-of-the-art approaches such as G-ODIN (Hsu et al., 2020), VOS (Du et al., 2022) and NDA (Sinha et al., 2021) can fail under a variety of open-set recognition settings, the proposed approach consistently leads to high-fidelity OOD detectors, without compromising the test accuracy.

translation (Wang et al., 2017) or compositional strategies such as Augmix (Hendrycks et al., 2020), TrivialAug (Müller and Hutter, 2021), Augmax (Wang et al., 2021), ALT (Gokhale et al., 2022), etc. On the other hand, Outlier Exposure (OE) (Hendrycks et al., 2018) that enforces the detector to flag samples from a carefully curated OOD dataset is the *modus operandi* for outlier specification. In practice, however, it is non-trivial to construct such outlier datasets for calibration. Hence, generating synthetic outliers in lieu of explicit curation is a more practical alternative. For example, Du *et al.* (Du et al., 2022) proposed Virtual Outlier Synthesis (VOS) which synthesizes outliers in the latent space of a classifier while Sinha *et al.* (Sinha et al., 2021) create pixel-space outliers with the aid of generative models. While these techniques have been shown to be effective for natural image benchmarks, we make an important finding that they are ineffective in medical imaging. As shown in Figure 1, existing inlier/outlier specification protocols do not lend themselves to practical open-set recognition settings, namely novel class and modality shift detection.

In this paper, we posit that the space in which the inlier and outlier augmentations are synthesized plays a central role in improving the performance of medical OOD detectors. Using an energy-based (Liu et al., 2020) training framework and an extensive empirical study,

we show that latent-space inlier synthesis coupled with diverse, pixel-space outlier synthesis consistently lead to high fidelity detectors. Note, our approach is straightforward to integrate with any prediction task or OOD scoring mechanism, and can produce significantly improved models for open-set recognition problems. Our codes will be made publicly available[1].

## 2. Problem Setup

**Setup.** We consider a $K-$way classifier $\mathcal{F}_\theta$ trained using labeled data $\mathcal{D} = \{(\mathbf{x}_i, y_i)\}_{i=1}^M$, where $\mathbf{x}_i$ is an image drawn from $P_{\mathrm{ID}}(\mathrm{x})$, and $y_i \in \mathcal{Y}_{\mathrm{ID}} = \{1, 2, \cdots, K\}$ is its corresponding label. The goal of OOD detection is to flag samples $\bar{\mathbf{x}} \in P_{\mathrm{OOD}}(\mathrm{x})$ that may correspond to covariate or semantic shifts with respect to $P_{\mathrm{ID}}(\mathrm{x})$. In this paper, we consider the challenging setting of open-set recognition, where the OOD data comes from classes that were not observed during training, *i.e.*, $\mathcal{Y}_{\mathrm{OOD}} \neq \mathcal{Y}_{\mathrm{ID}}$. This encompasses two broad categories: (a) *Novel classes*: In this case, the OOD data comes from the same imaging modality as the training set, but corresponds to a class unseen during the training phase (such as new diseases or healthy control groups); (b) *Modality shifts*: This refers to scenarios where the OOD images arise from disparate image modalities or organs, thus presenting completely unrelated semantic concepts. In practice, this is known to be significantly challenging, given the need to handle the diversity in the OOD set, and the propensity of deep models to associate these semantically unrelated images into one of the observed classes.

**OOD Detector Design.** A variety of OOD detection frameworks currently exist in the literature, ranging from energy-based (Liu et al., 2020) to density-based (Lee et al., 2018; Morningstar et al., 2021) and constrastively trained detectors (Sehwag et al., 2021; Tack et al., 2020). While our approach does not make any assumptions and can be used with any detector, we focus on energy based detectors and margin-based calibration in this work, which continue to be highly competitive in vision applications (Yang et al., 2021b). The free energy function for discriminative models (Liu et al., 2020) maps an input $\mathbf{x}$ to a deterministic scalar $E(\mathbf{x}; \theta)$ that is linearly aligned with log-likelihood $\log(P_{\mathrm{ID}}(\mathrm{x}))$. Mathematically, $E(\mathbf{x}; \theta) = -T \log \sum_{k=1}^K \exp \mathcal{F}_\theta^k(\mathbf{x})/T$, where $\mathcal{F}_\theta^k$ denotes the logit for class $k$ and $T$ is the temperature scaling parameter. We adopt the energy function to train an OOD detector $G$ alongside the classifier, similar to (Liu et al., 2020) where $G$ is defined as,

$$G(\mathbf{x}; \theta, \tau) = \begin{cases} \text{outlier}, & \text{if } -E(\mathbf{x}; \theta) \leq \tau, \\ \text{inlier}, & \text{if } -E(\mathbf{x}; \theta) > \tau. \end{cases} \quad (1)$$

Here, $\tau$ is a user-defined threshold for detection. Since the training data is expected to be characterized by low energy in comparison to OOD, we use negative energy scores to align with the notion that ID samples should have higher scores over OOD samples.

In practice, it is important to calibrate $G$ such that the dual objective of not compromising ID performance and reliably flagging OOD data are met. This can be formally stated as:

$$\min_\theta \mathop{\mathbb{E}}_{(\mathbf{x}, y) \in \mathcal{D}} \mathcal{L}_{CE}(\mathcal{F}_\theta(\mathbf{x}), y) + \alpha \mathop{\mathbb{E}}_{\tilde{\mathbf{x}} \in \mathcal{D}_{\mathrm{in}}} \mathcal{L}_{\mathrm{ID}}(E(\tilde{\mathbf{x}}); \theta) + \beta \mathop{\mathbb{E}}_{\bar{\mathbf{x}} \in \mathcal{D}_{\mathrm{out}}} \mathcal{L}_{\mathrm{OOD}}(E(\bar{\mathbf{x}}); \theta). \quad (2)$$

---

1. https://github.com/LLNL/OODmedic

Here, $\mathcal{L}_{CE}(.)$ is the standard cross-entropy loss. The terms $\mathcal{L}_{\text{ID}}$ and $\mathcal{L}_{\text{OOD}}$ (implemented as margin losses) are used to calibrate the OOD detector to operate as expected in the regimes of the specified inliers ($\mathcal{D}_{\text{in}}$) and outliers ($\mathcal{D}_{\text{out}}$). The success of this optimization hinges on the appropriate specification of inliers and outliers, which is the focus of this work.

## 3. Approach: Calibrating OOD Detectors

We study the implementation of (2) by exploring choices for inlier and outlier specification. In this context, we focus on the use of synthetic augmentations, without requiring additional data curation or explicit flagging (human supervision) of OOD data.

### 3.1. Augmentations for Inlier Synthesis

A popular strategy for improving the generalization of classifier models is to leverage data augmentation strategies. While it is common to utilize pixel-space transformations, we propose to leverage latent-space augmentations as an alternative choice for inlier synthesis.

***Pixel-space Synthesis.*** In this case, inliers are generated directly in the pixel-space by leveraging known statistical invariances. Following state-of-the-art, we consider the following strategies to perform inlier synthesis:- (i) conventional image manipulations such as random horizontal, vertical flips, or color jitter; and (ii) compositional strategies such as Augmix (Hendrycks et al., 2020) that synthesize inliers as a composition of multiple geometric and perceptual transformations.

***Latent-space Synthesis.*** While pixel-space augmentations are known to often aid the classifier performance, it is possible that they may adversely impact model safety (Hendrycks et al., 2021), e.g., outlier detection or calibration under real-world shifts, due to over-generalization. In order to systematically calibrate OOD detectors, while also controlling the risk of over-generalization, we propose to synthesize inliers in the low-dimensional latent space of a classifier. Formally, we assume that the model $\mathcal{F}$ can be decomposed into feature extractor and classifier modules as $\mathcal{F} = h \circ c$, and we approximate data from class $k$ in the feature space as $p(h(\mathbf{x})|y = k) \sim \mathcal{N}(\widehat{\boldsymbol{\mu}}_k, \widehat{\boldsymbol{\Sigma}})$. Each class is modeled using a class-specific mean $\widehat{\boldsymbol{\mu}}_k \in \mathbb{R}^d$ and a shared covariance $\widehat{\boldsymbol{\Sigma}} \in \mathbb{R}^{d \times d}$. Here, $d$ denotes the latent feature dimension and the class-specific statistics are obtained via maximum likelihood estimation. In order to synthesize class-specific inliers, we sample each of the $K$ gaussians from regions of low-likelihood corresponding to the tails as follows: $\mathcal{T} = \{\mathbf{t}_k | \mathcal{N}(\widehat{\boldsymbol{\mu}}_k, \widehat{\boldsymbol{\Sigma}}) < \delta\}_{k=1}^K$. Here $\mathbf{t}_k$ denotes the inlier sampled from the $k^{th}$ gaussian distribution. The modeling of class-specific gaussian distributions with a tied covariance allows the predictive model to be viewed under the lens of linear discriminant analysis (LDA) (Lee et al., 2018). If $p(y|h(\mathbf{x}))$ denotes the inferred posterior label distribution, we have,

$$p(y = c|h(\mathbf{x})) = \frac{\exp\left(\widehat{\boldsymbol{\mu}}_c^\top \widehat{\boldsymbol{\Sigma}}^{-1} h(\mathbf{x}) - \frac{1}{2}\widehat{\boldsymbol{\mu}}_c^\top \widehat{\boldsymbol{\Sigma}}^{-1} \widehat{\boldsymbol{\mu}}_c + \log\beta_c\right)}{\sum_{k=1}^K \exp\left(\widehat{\boldsymbol{\mu}}_k^\top \widehat{\boldsymbol{\Sigma}}^{-1} h(\mathbf{x}) - \frac{1}{2}\widehat{\boldsymbol{\mu}}_k^\top \widehat{\boldsymbol{\Sigma}}^{-1} \widehat{\boldsymbol{\mu}}_k + \log\beta_k\right)}, \tag{3}$$

where $\beta_c$ denotes the prior probabilities. On comparing (3) with the standard softmax based prediction as well as with the definition of energy, we observe that $E(\mathbf{x}, y = c) = -\widehat{\boldsymbol{\mu}}_c^\top \widehat{\boldsymbol{\Sigma}}^{-1} h(\mathbf{x}) + \frac{1}{2}\widehat{\boldsymbol{\mu}}_c^T \Sigma^{-1} \widehat{\boldsymbol{\mu}}_c - \log\beta_c$. Invoking the definition of the Gaussian density function, and by expressing kernel parameters in terms of energy, we can relate the energy scores for the latent space mean $\widehat{\boldsymbol{\mu}}_k$ and the tail $\mathbf{t}_k$ as

$$E\left(h(\mathbf{x}) = \widehat{\boldsymbol{\mu}}_k, y = k\right) - E\left(h(\mathbf{x}) = \mathbf{t}_k, y = k\right) < \frac{1}{2}(\mathbf{t}_k - \widehat{\boldsymbol{\mu}}_k)^\top \widehat{\boldsymbol{\Sigma}}^{-1}(\mathbf{t}_k + \widehat{\boldsymbol{\mu}}_k). \quad (4)$$

In particular, we obtain (4) from (3) using the fact the probability density of a Gaussian at its mean is greater the density at the tail and rearranging the obtained terms. For simplicity, we reuse the same notation $E$ to define the energy for $\mathbf{x} \in \mathcal{D}$ or equivalently $h(\mathbf{x})$ in the latent space. We find that the free energy $E(h(\mathbf{x}) = \mathbf{t}_k)$ can be bounded as:

$$E(h(\mathbf{x}) = \mathbf{t}_k) > -\log \sum_{k=1}^{K} \exp\left( - E(h(\mathbf{x}) = \widehat{\boldsymbol{\mu}}_k, k) + \frac{1}{2}(\mathbf{t}_k - \widehat{\boldsymbol{\mu}}_k)^\top \widehat{\boldsymbol{\Sigma}}^{-1}(\mathbf{t}_k + \widehat{\boldsymbol{\mu}}_k)\right) \quad (5)$$

Our optimization in (2) attempts to minimize the free energy for the inlier samples $\mathbf{t}_k$. From the expression (5), it becomes apparent that the model is encouraged to minimize the term $(\mathbf{t}_k - \widehat{\boldsymbol{\mu}}_k)$, *i.e.*, push the tail samples closer to the class-specific means and thereby improve generalization beyond the prototypical samples. When compared to pixel-space inliers, latent-space inliers include more challenging examples, albeit with reduced diversity. While we use the class-conditional Gaussian assumption similar to Mahalanobis distance-based detectors (Lee et al., 2018), note that, we utilize it for inlier synthesis and not for designing the detector itself. From our empirical study, we find that, when combined with an appropriate outlier specification, this leads to significant improvements in both novel class and modality shift detection without requiring any additional dataset-specific tuning.

## 3.2. Augmentations for Outlier Synthesis

In addition to inlier specification, exposure to representative outliers (Hendrycks et al., 2018; Roy et al., 2022; Thulasidasan et al., 2021; Sinha et al., 2021; Zhang et al., 2021; Chen et al., 2021) is critical to calibrate OOD detectors. Since carefully curated, diverse outlier datasets are not always available, we resort to generating synthetic outliers.

***Latent-space Synthesis.*** Following (Du et al., 2022), we can synthesize latent-space outliers as tail samples from class-specific gaussians in the penultimate layer of a classifier. During model training, we enforce such samples to be associated with maximum free energy.

***Pixel-space Synthesis.*** We construct pixel-space outliers as a set of severely corrupted versions of training samples. This is motivated by the need for exposing models to rich outlier data, so that the OOD detector can be calibrated to handle a variety of OOD scenarios. In contrast to latent-space outliers, pixel-space outliers distort the global features of the ID data and produce statistically disparate examples. In our implementation, we consider two augmentation strategies, where one of them is randomly chosen in every iteration: (i) `Augmix o Jigsaw`: We first transform an image using Augmix (Hendrycks et al., 2020) with high severity (set to 11), and subsequently distort using the Jigsaw corruption (divide an

image into 16 patches and perform patch permutation); (ii) `RandConv` (Xu et al., 2021): We used random convolutions with very large kernel sizes (chosen from $9-19$) to produce severely corrupted versions of the training images. We find that the inherent diversity of this outlier construction consistently leads to large performance gains, in particular for modality shift detection, in comparison to latent-space outliers which offer limited diversity.

### 3.3. Training

We define the loss functions in (2) as follows to implement our approach.

$$\mathcal{L}_{\mathrm{ID}} = \left[\max\left(0, E(h(\mathbf{x}) = \mathbf{t}_k) - m_{\mathrm{ID}}\right)\right]^2; \mathcal{L}_{\mathrm{OOD}} = \left[\max\left(0, m_{\mathrm{OOD}} - E(\mathbf{x} = \bar{\mathbf{x}})\right)\right]^2.$$

Here, $\mathcal{L}_{\mathrm{ID}}$ is a margin based loss with margin parameter $m_{\mathrm{ID}}$ for minimizing the energy $E(.)$ of the synthesized inliers. Similarly, for the outlier data, we define $\mathcal{L}_{\mathrm{OOD}}$ with margin parameter $m_{\mathrm{OOD}}$, so that the energy for those samples is maximized. Note, the losses can be suitably modified for the different inlier/outlier specification. For all experiments, we used the default hyper-parameters obtained using the higher-resolution ISIC2019 dataset, namely $m_{\mathrm{ID}} = -20, m_{\mathrm{OOD}} = -7, \alpha = \beta = 0.1$. Note, all hyper-parameters were chosen to maximize the validation (balanced) accuracy, since that is a metric that can be used when we assume no access to the OOD settings during model training.

## 4. Experiments

**Setup.** We use a large suite of medical imaging benchmarks and different model architectures to evaluate our approach in open-set recognition[2].

MedMNIST (Yang et al., 2021a) is a biomedical image corpus containing different imaging modalities, with all images pre-processed into size $28 \times 28$. In this study, we consider the following datasets from the corpus: (i) Blood MNIST, (ii) Path MNIST, (iii) Derma MNIST (iv) Oct MNIST, (v) Tissue MNIST and (vi)-(viii) Organ(A,C,S) MNIST.

ISIC2019 Skin Lesion Dataset (Tschandl et al., 2018; Codella et al., 2018; Combalia et al., 2019) is a skin lesion classification dataset containing a total of $25,331$ images belonging to 8 disease states namely Melanoma (MEL), Melanocytic nevus (NV), Basal cell carcinoma (BCC), Actinic keratosis (AK), Benign keratosis (BKL), Dermatofibroma (DF), Vascular lesion (VASC) and Squamous cell carcinoma (SCC). All images were resized to $224 \times 224$ as a preprocessing step.

NCT (Colorectal Cancer) (Kather et al., 2018) contains $100,000$ examples of $224 \times 224$ histopathology images of colorectal cancer and normal tissues from 9 possible categories namely, Adipose (ADI), background (BACK), debris (DEB), lymphocytes (LYM), mucus (MUC), smooth muscle (MUS), normal colon mucosa (NORM), cancer-associated stroma (STR), colorectal adenocarcinoma epithelicum (TUM).

**Model Architectures.** For all experiments with the MedMNIST benchmark, we resize the images to $32 \times 32$ and utilize the $40-2$ WideResNet architecture (Zagoruyko and Komodakis, 2016). To understand the generality of our method across different deep models,

---

2. The details of the benchmarks and experiment settings used can be found in the appendix

for experiments on ISIC2019 and NCT, we employ the ResNet-50 (He et al., 2016) model pre-trained on ImageNet (Deng et al., 2009).

**Evaluation Metrics.** (i) Balanced Validation Accuracy; (i) Area Under the Receiver Operator Characteristic curve (AUROC), a threshold independent metric, that reflects the probability that an in-distribution image is assigned a higher confidence over the OOD samples and (iii) Area under the Precision-Recall curve (AUPRC) where the ID and OOD samples are considered as positives and negatives respectively (included in the supplement).

**Training Protocols.** We compare the proposed inlier/outlier specification with the following state-of-the-art approaches: (i) VOS: This method uses latent-space outliers from (Du et al., 2022); (ii) VOS++: In this variant, we combine the VOS latent-space outliers with pixel-space inliers generated using Augmix (Hendrycks et al., 2020); (iii) NDA: This method utilizes pixel-space outliers similar to (Sinha et al., 2021); and (iv) NDA++: This variant of NDA employs Augmix to generate pixel-space inliers in addition to the pixel-space outliers. Furthermore, we consider another outlier exposure-free baseline, Generalized ODIN (G-ODIN) (Hsu et al., 2020) as a representative for methods that do not employ any additional calibration to the model itself (only fine-tunes the noise parameter as a post-hoc step), and to highlight the fact that such a baseline can sometimes outperform even sophisticated approaches. Note, for all methods including ours, we fixed the model architecture, loss function and the training settings to be the same, in order to isolate the impact of the augmentation design.

### 4.1. Results

**Detecting Novel Classes.** In this setting, test samples may belong to new disease states or control group patients that were not observed during training. The subtle variations in image statistics across classes in medical images make detection of these out of distribution samples challenging. In our experiments, we held out a subset of classes for all benchmarks and presented them to the models at test time. The performance summary in Table 1 illustrates the novel class detection performance of different calibration strategies. It can be seen that detectors designed with our approach achieved the best performance across all datasets (gains of $15\% - 28\%$ on average). While the G-ODIN detector and VOS perform competitively in some cases, they exhibit large variance across benchmarks.

**Modality Shift Detection.** With the MedMNIST benchmark, we treated each dataset as ID and evaluated the out-of-distribution (OOD) detection performance on the remaining 7 datasets. We included details of the OOD data used to evaluate our models on the ISIC2019 and Colorectal Cancer benchmarks in the supplement. Table 1 summarizes the performance of different calibration protocols. As observed from the AUROC scores, our approach consistently outperforms all baselines by significant margins ($10 - 30\%$ on average), while maintaining generalizability to the ID test set (refer to Figure 6 in the supplement for the balanced accuracy scores, i.e., average of specificity and sensitivity). G-ODIN and even the state-of-the-art baselines, VOS and VOS++, underperform in this challenging setting, when compared to methods that utilize pixel-space outliers.

Table 1: **Performance evaluation**. Average AUROC for modality shift and novel class detection on all the benchmarks considered. In each case, we highlight the best and the second best performing methods in green and orange respectively. We refer the readers to Tables 4-12 and Figure 6 in the supplement for a fine-grained characterization of the performance of different methods.

| In Dist. | AUROC for Modality Shift/Semantic Novelty Detection | | | | | |
|---|---|---|---|---|---|---|
| | **G-ODIN** | **VOS** | **VOS++** | **NDA** | **NDA++** | **Ours** |
| Colorectal | 91.8 / 53.8 | 85.3 / 86.4 | 76.8 / 70.8 | 98.7 / 67.5 | 59.0 / 72.3 | 99.9 /94.2 |
| ISIC2019 | 70.9 / 65.7 | 54.9 / 72.3 | 78.0 / 75.6 | 80.7 / 71.7 | 79.5 / 75.3 | 97.1 / 83.1 |
| BloodMNIST | 88.7 / 53.9 | 89.4 / 44.7 | 84.2 / 38.2 | 96.2 / 66.0 | 95.8 / 53.5 | 99.7 / 89.1 |
| PathMNIST | 84.4 / 51.7 | 77.5 / 39.4 | 71.0 / 71.7 | 96.1 / 37.4 | 61.1 / 57.0 | 98.9 / 71.2 |
| DermaMNIST | 85.3 / 69.3 | 64.1 / 67.5 | 85.3 / 72.9 | 95.2 / 51.23 | 80.0 / 69.9 | 96.6 / 75.5 |
| OCTMNIST | 49.0 / 47.2 | 50.1 / 55.4 | 68.0 / 52.0 | 92.8 / 51.0 | 94.4 / 75.2 | 99.6 / 78.9 |
| TissueMNIST | 82.7 / 55.2 | 72.9 / 46.4 | 60.2 / 28.0 | 81.1 / 42.3 | 70.2 / 58.8 | 96.6 / 83.4 |
| OrganAMNIST | 95.8 / 89.9 | 73.7 / 62.2 | 77.8 / 73.9 | 70.2 / 44.4 | 96.2 / 75.6 | 99.7 / 98.1 |
| OrganSMNIST | 80.3 / 82.0 | 51.5 / 47.0 | 62.1 / 72.0 | 94.0 / 83.9 | 92.9 / 88.1 | 98.2 / 93.9 |
| OrganCMNIST | 85.7 / 79.3 | 56.6 / 58.8 | 64.6 / 65.17 | 93.2 / 83.8 | 94.2 / 81.5 | 99.1 / 97.5 |

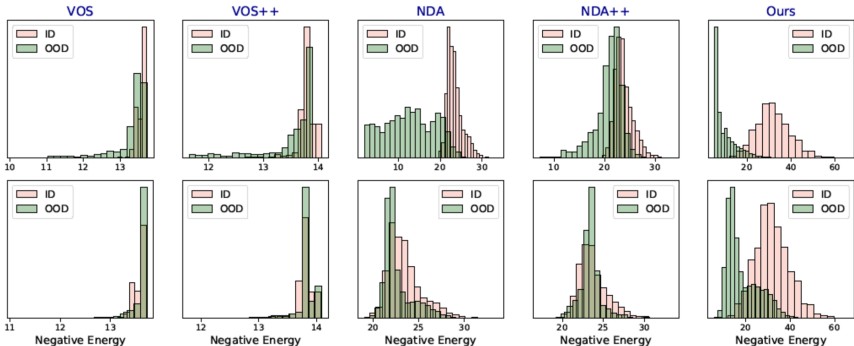

Figure 2: **Histograms of negative energy scores**. We plot the scores obtained using different inlier and outlier specifications. With BloodMNIST as ID, the top row corresponds to modality shifts (OOD: DermaMNIST) and the bottom row shows novel classes.

## 5. Discussion

From the empirical results in this study, we conclude that the space in which the inlier/outlier augmentations are specified plays a crucial role in effectively calibrating OOD detectors. Importantly, the inherent diversity offered by the pixel-space outlier synthesis is essential for handling modality shifts. This behavior is further emphasized by the observation that both NDA-based baselines outperform VOS approaches that synthesize latent-space outliers with

limited diversity. On the other hand, with novel class detection, we find that our approach which samples hard inliers in the latent space is particularly effective. Figure 2 depicts the histograms of the negative energy scores for the case of Blood MNIST (ID), wherein the modality shift results were obtained using DermaMNIST. We observe that our approach effectively distinguishes between ID and OOD distributions (much higher scores for ID data) in both cases, while the other approaches contain a high overlap. Overall, this study provides an optimal protocol to construct synthetic inliers/outliers for calibrating OOD detectors, and demonstrates state-of-the-art performance on open-set recognition.

## Acknowledgments

This work was performed under the auspices of the U.S. Department of Energy by the Lawrence Livermore National Laboratory under Contract No. DE-AC52-07NA27344, Lawrence Livermore National Security, LLC. and was supported by the LLNL-LDRD Program under Project No. 22-ERD-006. LLNL-CONF-835509.

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

## Appendix A. Related Work

**Out-of-Distribution detection.** This is the task of identifying whether a given sample is drawn from the in-distribution data manifold or not. Such a task requires an effective scoring metric that can distinguish between ID and OOD data. In this context, much of recent research has focused on designing useful scoring functions to improve detection over different regimes of OOD data. For instance, Hendrycks *et al.* (Hendrycks and Gimpel, 2017) proposed the Maximum Softmax Probability (MSP) score as a strong baseline for OOD detection. Subsequently, Liang *et al.* (Liang et al., 2018) proposed ODIN, a scoring function based on re-calibrating the softmax probabilities through temperature scaling and input pre-processing. On similar lines, Lee *et al.* (Lee et al., 2018) utilized Mahalanobis distances accumulated from the classifier latent spaces as a scoring metric. Ren *et al.* (Ren et al., 2021) proposed the relative mahalanobis distance as an effective score for fine-grained OOD detection. Sastry *et al.* (Sastry and Oore, 2020) proposed a latent space scoring metric for detecting outliers by comparing Gram matrices. More recently, Liu *et al.* (Liu et al., 2020) proposed to use the energy metric for OOD detection. The metric is directly related to the underlying data likelihood and is known to produce significantly improved OOD detectors. Owing to the ease of adoption and success of the energy metric in OOD detection, without loss of generality, we adopt energy as the scoring function in this paper.

**OE-free OOD Detection**. The objective defined in (2) requires the OOD detector to be calibrated with pre-specified, curated outlier data. However, it is significantly challenging to construct such datasets in practice, thus motivating the design of 'OE-Free' methods. With the requirement of the ODIN detector to be fine-tuned with pre-specified OOD data, Hsu *et al.*(Hsu et al., 2020) proposed Generalized ODIN (G-ODIN) as an outlier data-free variant of ODIN, while also improving the detection performance. On the other hand, Du *et al.* (Du et al., 2022) proposed to synthesize virtual outliers by sampling hard negative examples (i.e, samples at the class decision boundaries) directly in the latent space of a classifier to calibrate the OOD detector, in lieu of OOD calibration datasets. Our formulation broadly falls under the class of OE-free methods as we leverage only synthetic outliers.

## Appendix B. Dataset Descriptions

**MedMNIST Benchmark.** (i) Blood MNIST consists of $17,092$ human blood cell images collected from healthy individuals corresponding to 8 different classes; (ii) Path MNIST is a histology image dataset of colorectal cancer with $107,180$ samples of non-overlapping, hematoxylin and eosin stained image patches from 9 different classes; (iii) Derma MNIST is a skin lesion dataset curated from the HAM1000 (Tschandl *et al.*) database. It contains a total of $10,015$ images across 7 cancer types; (iv) Oct MNIST contains $109,309$ optical coherence

Table 2: **Known and Novel Classes selected from the MedMNIST Benchmark**

| Datasets | Blood | Path | Derma | OCT | Tissue | OrganA,C,S |
|---|---|---|---|---|---|---|
| **Known Classes** | $1-5,7$ | $1-5,7$ | $1,3-6$ | $1,2,4$ | $1-2,4-5,7-8$ | $1,5-11$ |
| **Novel Classes** | $6,8$ | $6,8,9$ | $2,7$ | $3$ | $3,6$ | $2,3,4$ |

Table 3: **Ablation Study on the Choice of Inlier/Outlier Specification on the ISIC2019 Benchmark.** We report the average AUROC (%) scores across modality shifts and semantic novelty detection.

| Pix. In | Lat. In | Pix. In + Pix. Out | Pix. In + Lat. Out | Lat In. + Pix. Out |
|---------|---------|--------------------|--------------------|--------------------|
| 71.3 | 72.2 | 77.3 | 61.5 | **91.8** |

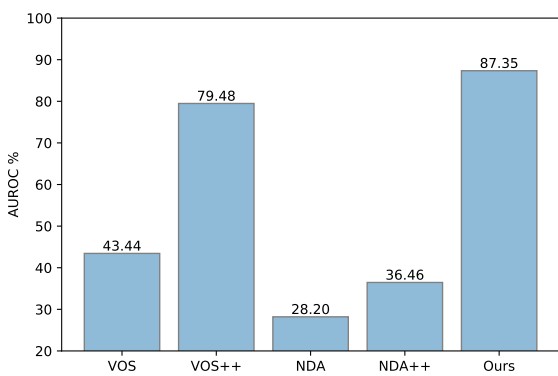

Figure 3: **Detecting Covariate Shifts (New hospital) on Camelyon-17**. We report the AUROC of different approaches trained with a Resnet-50 backbone.

tomography (OCT) retinal images corresponding to 4 diseases; (v) Tissue MNIST is a kidney cortex image dataset curated from the Broad Bioimage Benchmark Collection with $236,386$ images from 8 classes; (vi) (vii) (viii) Organ(A,C,S) MNIST are images of abdominal CT collected from the Axial, Coronal and Sagittal planes of 3D CT images from the Liver-tumor segmentation benchmark. The datasets contain $58,850$, $23,660$ and $25,221$ images across 11 classes respectively. For each of MedMNIST datasets, we consider the validation splits from all remaining datasets for evaluating modality shift detection performance. On the other hand, we use a subset of classes held-out during training to evaluate the novel class detection.

**ISIC 2019 and NCT Datasets.** For these two benchmarks, the following datasets were used to evaluate OOD detection performance. For each OOD dataset, we highlight if its a modality shift (M) or a semantic shift/novel class (S) along with the corresponding ID dataset (ISIC or NCT) with which the model was trained:- (i) Camelyon-17 (WILDS) (Bandi et al., 2018)(M:ISIC, S: NCT) is a histopathology dataset of tumor and non-tumor breast cells with approximately 450K images curated from five different medical centers. We randomly sample 3000 examples from the dataset for OOD detection; (ii) Knee (M: ISIC, M: NCT) Osteoarthritis severity grading dataset contains X-ray images of knee joints with examples corresponding to arthritis progression. We used 825 examples chosen randomly from the dataset for evaluation; (iii) CXR (M: ISIC, M: NCT)[3] is a chest X-ray dataset curated from

---

3. https://github.com/cxr-eye-gaze/eye-gaze-dataset

the MIMIC-CXR database containing $1,083$ samples corresponding to disease states namely normal, pneumonia and congestive heart failure and (iv) Retina (M: ISIC, M: NCT) is a set of 1500 randomly chosen retinal images with different disease progressions from the Diabetic Retinopathy detection benchmark from Kaggle[4]; (v) Clin Skin (S: ISIC) contains 723 images of healthy skin (Pacheco et al., 2020); (vi) Derm-Skin (S: ISIC) consists of 1565 dermoscopy skin images obtained by randomly cropping patches in the ISIC2019 database (Pacheco et al., 2020); (vii) NCT 7K (S: NCT) contains 1350 histopathology images of colorectal adenocarcinoma with no overlap with NCT (Pacheco et al., 2020). In addition, we use 2000 randomly chosen examples from ISIC as a source of modality shift for the detector trained on NCT and vice-versa. Moreover, in both cases, novel classes unseen while training are also used to evaluate detection under semantic shifts.

Table 4: **AUPR scores for novel class detection on the MedMNIST benchmark.** We report the AUPR (Input) scores using different approaches with a $40 - 2$ WideResNet backbone.

| In Dist. | Methods | | | | | |
|---|---|---|---|---|---|---|
| | **G-ODIN** | **VOS** | **VOS++** | **NDA** | **NDA++** | **Ours** |
| Blood | 47.89 | 25.02 | 21.98 | 37.44 | 35.72 | 73.32 |
| Path | 20.29 | 12.33 | 36.23 | 11.74 | 26.61 | 30.35 |
| Derma | 79.36 | 75.18 | 82.4 | 57.58 | 80.69 | 82.28 |
| OCT | 53.56 | 57.35 | 63.1 | 61.39 | 76.62 | 79.45 |
| Tissue | 55.53 | 48.37 | 39.75 | 55.36 | 66.96 | 87.79 |
| OrganA | 86.82 | 42.48 | 51.72 | 35.7 | 68.22 | 95.51 |
| OrganS | 72.61 | 31.99 | 51.81 | 72.56 | 81.69 | 89.32 |
| OrganC | 73.48 | 38.95 | 47.06 | 68.08 | 71.79 | 95.16 |

Table 5: **Full results for modality shift detection on the BloodMNIST dataset (ID) using the $40 - 2$ WideResnet (AUROC/AUPR metrics)**

| OOD Data | Methods | | | | | |
|---|---|---|---|---|---|---|
| | **G-ODIN** | **VOS** | **VOS++** | **NDA** | **NDA++** | **Ours** |
| Path | 88.0/57.4 | 71.6/38.8 | 67.9/33.2 | 75.7/36.2 | 97.6/82 | 99.0/96.5 |
| Derma | 89.3/73.8 | 69.6/68.8 | 70.1/78.2 | 97.4/97.3 | 83.6/86.2 | 98.8/99.1 |
| OCT | 96.7/89.3 | 98.2/95.1 | 95.8/82.6 | 100/100.0 | 95.8/77.3 | 100.0/99.9 |
| Tissue | 98.8/73.4 | 99.2/98.8 | 98.4/89.6 | 100/100.0 | 98.9/93.6 | 100/100.0 |
| OrganA | 99.4/84.0 | 95.9/89.3 | 86.7/71.4 | 100/100.0 | 98.2/94.3 | 100.0/99.9 |
| OrganC | 99.3/90.6 | 96.1/95.0 | 84.8/81.4 | 100/100.0 | 98.2/97.2 | 100.0/99.9 |
| OrganS | 99.5/92.2 | 95.5/93.8 | 85.6/81.6 | 100/100.0 | 98.6/97.7 | 100.0/99.9 |

---

4. https://www.kaggle.com/competitions/diabetic-retinopathy-detection/data

Table 6: **Full results for modality shift detection on the PathMNIST dataset (ID) using the $40-2$ WideResnet (AUROC/AUPR metrics)**

| OOD Data | Methods | | | | | |
|---|---|---|---|---|---|---|
| | **G-ODIN** | **VOS** | **VOS++** | **NDA** | **NDA++** | **Ours** |
| Blood | 92.1/94.4 | 99.3/99.7 | 79.4/92.8 | 89.7/97.7 | 64.9/91.0 | 95.3/99.2 |
| Derma | 72.8/95.7 | 74.3/94.7 | 39.8/85.1 | 91.0/98.2 | 43.6/87.4 | 98.1/99.7 |
| OCT | 91.8/95.9 | 69.2/70.3 | 86.6/84.3 | 98.1/97.8 | 79.7/81.8 | 99.7/99.6 |
| Tissue | 73.4/96.4 | 67.7/66.0 | 72.0/70.8 | 95.8/93.0 | 71.5/63.6 | 100.0/99.9 |
| OrganA | 76.7/83.4 | 72.9/75.9 | 72/66.7 | 99.6/99.7 | 52.7/60.7 | 100.0/100.0 |
| OrganC | 76.0/92.7 | 78.0/89.9 | 71.5/85.4 | 99.4/99.8 | 56.7/83.4 | 99.8/99.9 |
| OrganS | 75.1/90.9 | 81.4/91.4 | 75.4/86.5 | 99.4/99.8 | 58.4/83.4 | 99.8/99.9 |

Table 7: **Full results for modality shift detection on the DermaMNIST dataset (ID) using the $40-2$ WideResnet (AUROC/AUPR metrics)**

| OOD Data | Methods | | | | | |
|---|---|---|---|---|---|---|
| | **G-ODIN** | **VOS** | **VOS++** | **NDA** | **NDA++** | **Ours** |
| Blood | 87.4/86.9 | 77.0/77.6 | 90.0/88.0 | 85.2/71.1 | 80.7/80.1 | 91.8/89.9 |
| Path | 82.2/66.1 | 72.7/44.5 | 82.7/57.9 | 90.4/46.2 | 92.5/72.0 | 89.6/56.1 |
| OCT | 79.0/49.8 | 72.6/49.0 | 81.4/73.6 | 100.0/99.5 | 55.1/35.4 | 99.5/95.4 |
| Tissue | 57.2/43.9 | 84.2/52.2 | 87.3/66.7 | 99.9/96.7 | 78.3/54.9 | 99.8/96.4 |
| OrganA | 79.2/76.2 | 49.9/20.7 | 85.9/73.5 | 97.3/83.6 | 85.3/66.5 | 98.2/90.2 |
| OrganC | 78.2/85.5 | 47.4/33.1 | 85.0/85.0 | 96.9/90.8 | 83.8/76.3 | 98.5/96.2 |
| OrganS | 78.2/85.4 | 44.8/31.7 | 85.2/80.7 | 96.6/90.3 | 84.0/76.4 | 99.1/98 |

## Appendix C. Ablation Study

In addition to the existing baseline methods, we performed an ablation study on the ISIC-2019 benchmark to compare other choices of inlier and outlier specification. As showed in Table 3, we observe that the inclusion of the latent space inliers (Lat. In) alone during the optimization is not sufficient to obtain high quality OOD detectors. This is due to the fact that optimizing to minimize the energy scores for the latent inliers in fact leads to over-generalization and cannot effectively demarcate decision boundaries between inliers and outliers. In practice, such over-generalization can even affect ID accuracy. In order to circumvent this issue, we propose the use of diverse, pixel-space outlier samples. Moreover, as argued in the paper, (i) using pixel-space inliers without any outlier synthesis leads to inferior OOD detection performance, and (ii) pixel-space inliers + pixel-space outliers is consistently better than pixel-space inliers+ Latent-space outliers, further emphasizing the effectiveness of diverse pixel-space outliers. Finally, synthesizing inliers and outliers from the latent space is not feasible, since both approaches sample from the tails of the class-conditioned distributions.

Table 8: **Full results for modality shift detection on the OctMNIST dataset (ID) using the $40-2$ WideResnet (AUROC/AUPR metrics)**

| OOD Data | Methods | | | | | |
|---|---|---|---|---|---|---|
| | **G-ODIN** | **VOS** | **VOS++** | **NDA** | **NDA++** | **Ours** |
| Blood | 97.4/94.7 | 60.5/91.2 | 75.2/96.1 | 97.5/99.5 | 97.0/99.6 | 100/100 |
| Path | 99.1/68.4 | 49.0/55.0 | 67.4/81.7 | 98.3/98.4 | 96.8/97.6 | 100/100.0 |
| Derma | 96.4/99.3 | 52.1/89.2 | 66.9/96.4 | 99.7/100.0 | 99.4/99.9 | 100/100 |
| Tissue | 78.0/42.3 | 40.9/25.4 | 62.0/64.2 | 54.8/40.8 | 90.9/80.8 | 97.5/95.0 |
| OrganA | 97.3/48.4 | 50.3/67.4 | 70.1/86.3 | 99.8/99.8 | 88.2/92.4 | 99.9/99.9 |
| OrganC | 98.2/72.9 | 48.2/81.2 | 66.8/92.6 | 99.7/99.9 | 94.1/98.5 | 100.0/100.0 |
| OrganS | 98.3/70.9 | 49.8/80.6 | 67.5/92.4 | 99.8/99.9 | 94.3/98.6 | 100.0/100.0 |

Table 9: **Full results for modality shift detection on the TissueMNIST dataset (ID) using the $40-2$ WideResnet (AUROC/AUPR metrics)**

| OOD Data | Methods | | | | | |
|---|---|---|---|---|---|---|
| | **G-ODIN** | **VOS** | **VOS++** | **NDA** | **NDA++** | **Ours** |
| Blood | 93.9/99.4 | 68.0/97.8 | 54.9/95.9 | 100/100 | 99.6/100.0 | 100/100 |
| Path | 93.8/98.0 | 83.7/95.3 | 64.4/87.1 | 99.9/100.0 | 97.7/99.0 | 100/100 |
| Derma | 91.0/99.1 | 84.3/99.2 | 74.9/98.5 | 99.2/100.0 | 98.9/99.9 | 100.0/100 |
| OCT | 20.9/54.9 | 46.8/75.2 | 25.0/64.0 | 13.7/49.8 | 11/49.0 | 77.6/89.5 |
| OrganA | 97.7/99.5 | 75.5/91.8 | 69.5/90.6 | 86.1/94.4 | 63.3/88.7 | 99.6/99.9 |
| OrganC | 97.1/99.6 | 76.4/97 | 67.2/95.7 | 84.3/97.3 | 62.2/94.9 | 99.6/100.0 |
| OrganS | 97.2/99.6 | 75.3/96.6 | 65.3/95.2 | 84.4/97.2 | 58.7/94.1 | 99.6/99.9 |

## Appendix D. Additional Results for Camelyon-17 - WILDS

In this study, we demonstrate the efficacy of our approach in detecting real-world covariate shifts on the WILDS benchmark (Bandi et al., 2018) curated from different hospitals. Follow-

Table 10: **Full results for modality shift detection on the OrganAMNIST dataset (ID) using the $40-2$ WideResnet (AUROC/AUPR metrics)**

| OOD Data | Methods | | | | | |
|---|---|---|---|---|---|---|
| | **G-ODIN** | **VOS** | **VOS++** | **NDA** | **NDA++** | **Ours** |
| Blood | 99.4/98.2 | 64.7/79.8 | 65.7/81.7 | 32.4/72.7 | 100.0/100.0 | 100.0/100.0 |
| Path | 99.5/91.4 | 67.7/49.8 | 67.3/50.8 | 83.1/66.3 | 99.4/98.8 | 99.2/98.9 |
| Derma | 96.4/99.4 | 76.9/89.5 | 76.0/91.2 | 78.5/92.1 | 99.6/99.9 | 99.4/99.9 |
| OCT | 88.2/98.4 | 63.9/37.2 | 92.4/74.8 | 70.6/47.1 | 93.2/88.1 | 99.9/99.8 |
| Tissue | 94.4/90.6 | 95.2/64.3 | 87.5/45.3 | 86.3/41.8 | 88.8/66.2 | 100/100.0 |

Table 11: **Full results for modality shift detection on the OrganSMNIST dataset (ID) using the $40 - 2$ WideResnet (AUROC/AUPR metrics)**

| OOD Data | Methods | | | | | |
|---|---|---|---|---|---|---|
| | **G-ODIN** | **VOS** | **VOS++** | **NDA** | **NDA++** | **Ours** |
| Blood | 92.8/64.2 | 49.4/66.9 | 64.7/77.2 | 91.6/93.8 | 100.0/100.0 | 99.9/99.9 |
| Path | 97.2/34.1 | 41.3/23.4 | 55.5/40.1 | 90.7/68.5 | 97.4/90.5 | 99.0/97.1 |
| Derma | 92.5/98.3 | 34.6/59.2 | 70.1/76.0 | 97.9/98.6 | 100.0/100.0 | 99.2/99.6 |
| OCT | 80.3/99.1 | 52.8/21.9 | 46.1/25.0 | 90.6/80.4 | 72.6/39.4 | 92.8/77.4 |
| Tissue | 93.6/90.6 | 79.4/18.6 | 74.4/18.0 | 99.3/93.2 | 94.7/76.2 | 100.0/99.8 |

Table 12: **Full results for modality shift detection on the OrganCMNIST dataset (ID) using the $40 - 2$ WideResnet (AUROC/AUPR metrics)**

| OOD Data | Methods | | | | | |
|---|---|---|---|---|---|---|
| | **G-ODIN** | **VOS** | **VOS++** | **NDA** | **NDA++** | **Ours** |
| Blood | 96.6/89.8 | 56.0/63.6 | 41.0/60 | 86.6/88.8 | 100.0/100.0 | 99.6/99.8 |
| Path | 98.8/59.4 | 53.2/23.1 | 71.9/36.5 | 98.1/93.1 | 99.8/99.2 | 99.7/99.4 |
| Derma | 97.8/95.5 | 53.2/65.3 | 68.2/77.2 | 96.2/97.2 | 99.7/99.8 | 98.7/99.3 |
| OCT | 96.2/87.3 | 59.5/19.6 | 79.0/34.6 | 92.0/66.5 | 91.0/65.4 | 98.2/96.0 |
| Tissue | 83.9/66.1 | 61.0/10.3 | 62.5/9.8 | 93.4/48.9 | 80.6/30.3 | 99.4/97.4 |

ing standard practice, we consider images from hospital 5 as the OOD data characterizing covariate shift and train/validate detectors on images from all the remaining hospitals. Figure 3 illustrates the detection performance (AUROC) of the different methods in detecting covariate shifts. We can observe that our approach significantly outperforms the baselines producing an improvement of $\sim 7 - 35\%$ in terms of AUROC.

## Appendix E. Details of Experiments in the Main Paper

**Dataset Preprocessing.** We first split each of the datasets into two categories namely (i) data from classes known during training (*known classes*) and (ii) data from classes unknown while training (*novel classes*) where the latter constitutes OOD data with semantic shifts. The dataset from the former category is split in the ratio of $90 : 10$ for training and evaluating the predictive models. Table 2 provides the list of known and novel classes for the MedMNIST benchmark. In case of ISIC2019, we choose BKL, VASC and SCC as novel classes while MUC, BACK and NORM are chosen as novel classes for the NCT(Colorectal) benchmark.
**Training Details.**
Estimating Class-specific Means and Joint Covariance: We estimate the means and joint covariance via maximum likelihood estimation during training, similar to (Du et al., 2022). We employ $K$ queues each of size 1000 where each queue is filled during every iteration until their pre-specified capacities with the class specific latent embeddings (extracted from the

Table 13: **Evaluation on the ISIC2019 benchmark.** We report AUROC scores obtained with a ResNet-50 model trained on the ISIC2019 dataset. Note, we show results for both novel classes (blue), and modality shifts (red).

| OOD Data | Methods | | | | | |
|---|---|---|---|---|---|---|
| | **G-ODIN** | **VOS** | **VOS++** | **NDA** | **NDA++** | **Ours** |
| Novel Classes | 62.20 | 75.04 | 68.69 | 65.41 | 68.38 | 74.00 |
| Clin Skin | 62.93 | 61.33 | 78.80 | 67.06 | 72.01 | 81.55 |
| Derm Skin | 71.93 | 80.53 | 79.27 | 82.73 | 85.5 | 93.9 |
| Wilds | 65.78 | 66.71 | 57.15 | 83.69 | 85.29 | 99.77 |
| Colorectal | 77.08 | 32.02 | 81.27 | 71.33 | 78.84 | 98.58 |
| Knee | 66.50 | 23.02 | 83.47 | 89.25 | 94.73 | 94.08 |
| CXR | 74.32 | 76.80 | 80.93 | 83.18 | 62.08 | 96.94 |
| Retina | 71.10 | 76.33 | 87.39 | 76.04 | 76.65 | 95.86 |
| Avg. | 68.98 | 61.47 | 77.12 | 77.34 | 77.94 | **91.84** |

Table 14: **Evaluation on the colorectal cancer benchmark.** We report AUROC scores obtained with a ResNet-50 model trained on the the colorectal cancer dataset (Kather et al., 2018). Note, we show results for both novel classes (blue) and modality shift detection (red).

| OOD Data | Methods | | | | | |
|---|---|---|---|---|---|---|
| | **G-ODIN** | **VOS** | **VOS++** | **NDA** | **NDA++** | **Ours** |
| Novel Classes | 41.59 | 84.24 | 63.13 | 79.38 | 74.34 | 94.06 |
| NCT 7K | 76.02 | 78.92 | 62.04 | 80.46 | 63.25 | 96.11 |
| WILDS | 43.82 | 95.97 | 87.31 | 42.73 | 79.4 | 92.47 |
| ISIC2019 | 79.03 | 65.6 | 85.46 | 98.71 | 65.17 | 99.86 |
| Knee | 95.55 | 95.26 | 58.87 | 96.63 | 44.67 | 99.98 |
| CXR | 95.99 | 99.19 | 67.18 | 99.79 | 71.65 | 99.91 |
| Retina | 96.67 | 81.06 | 95.62 | 99.68 | 54.66 | 100.0 |
| Avg. | 75.52 | 85.75 | 74.23 | 85.34 | 64.73 | **97.48** |

penultimate layer) of the training data. We then adopt an online strategy to update the queues such that they contain much higher quality embeddings of the data as the training progresses. In particular, we enqueue one class-specifc latent embedding to the respective queues while dequeuing one embedding from the same class.

Sampling the Latent Space: In practice, we select samples close to the class specific boundaries based on the $n^{th}$ smallest likelihood ($n = 64$) among $N$ examples ($N = 10,000$) synthesized from the respective Gaussian distributions.

General Hyperparameters: We train the $40 - 2$ WideResNet and ResNet-50 architectures for 100 and 50 epochs with learning rates of $1e{-}3$ and $1e{-}4$ respectively. We reduce the learning rate by a factor of 0.5 every 10 epochs using the Adam optimizer with a momentum of 0.9 and a weight decay of $5e^{-4}$. We choose a batch size of 128 for datasets from MedMNIST and

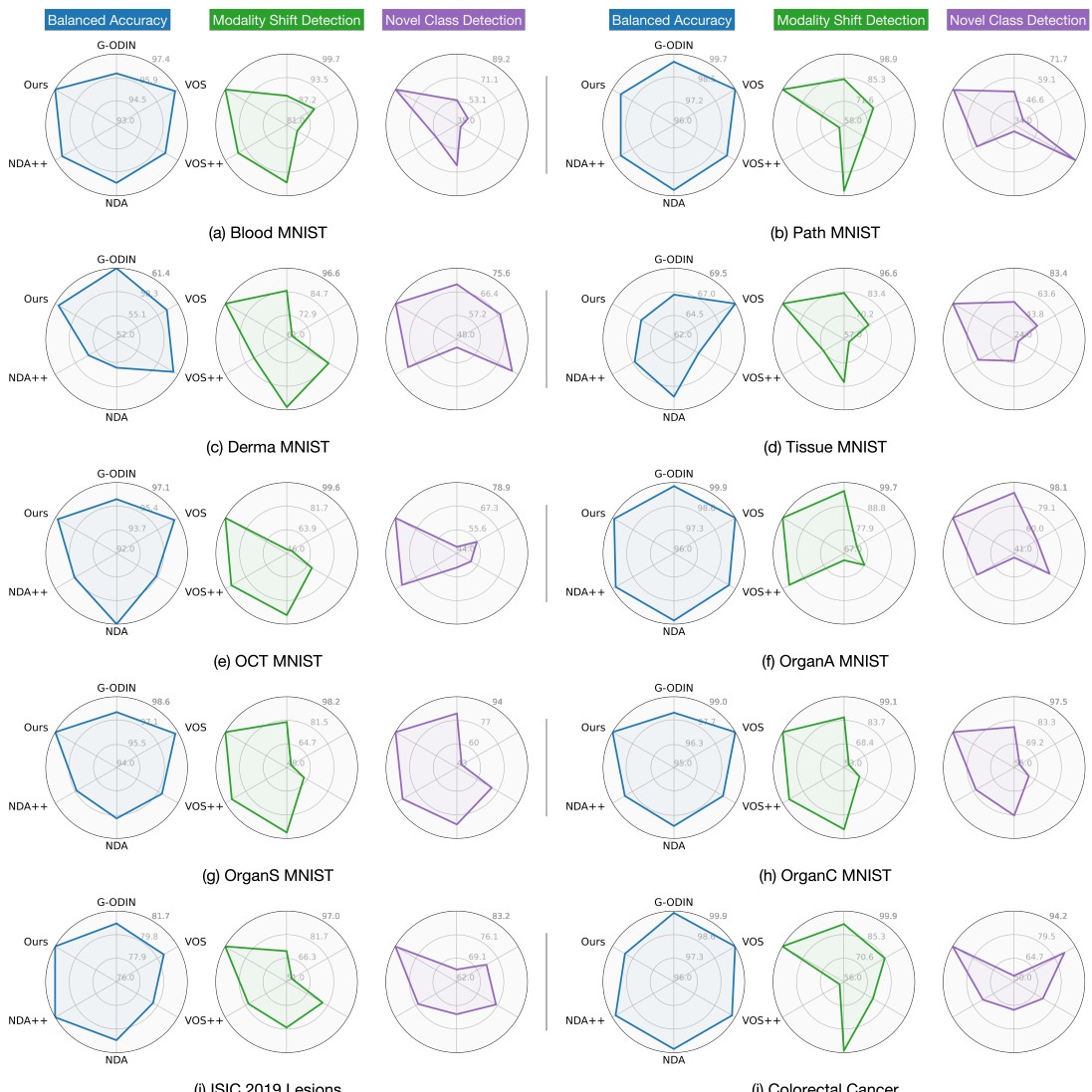

Figure 4: **Evaluation of OOD detectors calibrated with different inlier/outlier constructions.** The radar plots correspond to models trained on each of the benchmarks and they report the respective balanced test accuracy (%) (left), average AUROC (%) for modality shifts (middle) and novel class (right) detection.

64 for the full-sized images. For all experiments including the baselines (except G-ODIN), we use a margin $m_{\text{ID}} = -20$ and $m_{\text{OOD}} = -7$ with $\alpha = \beta = 0.1$. We introduce pixel-space synthetic outliers during the beginning of training for our approach and baselines except for the VOS variants where we introduce the outliers at epoch 40 following standard practice.

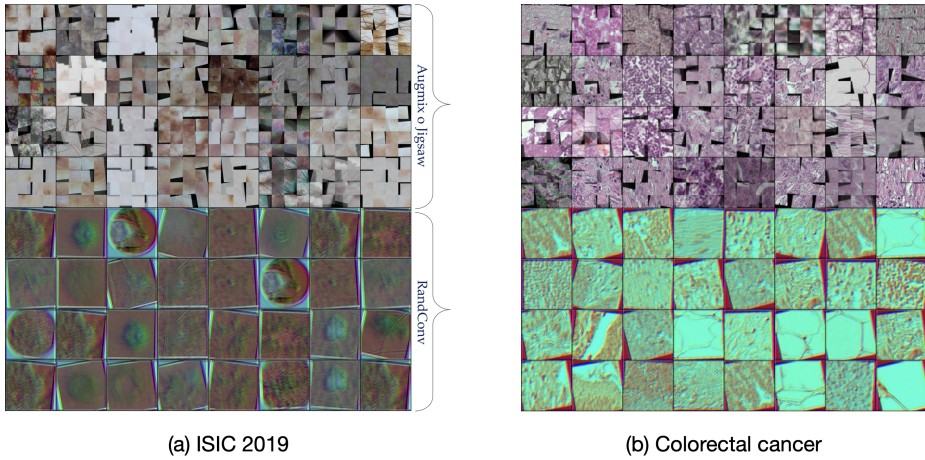

(a) ISIC 2019                    (b) Colorectal cancer

Figure 5: **Examples of Pixel-Space Synthetic Outliers.**

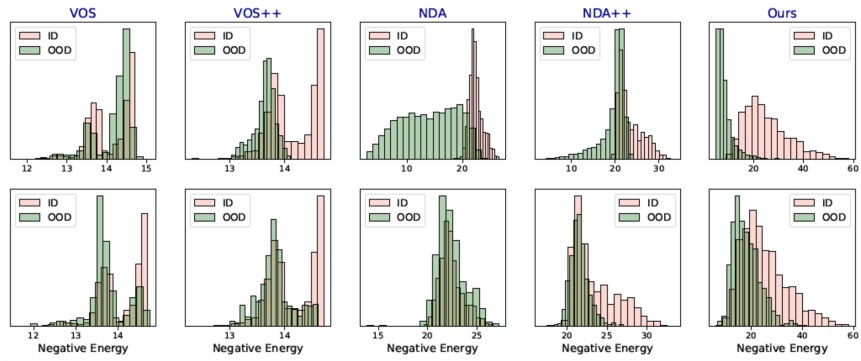

Figure 6: **Histograms of negative energy scores.** With DermaMNIST as ID, the top
row corresponds to modality shift (OOD: OrganAMNIST) and the bottom row
shows semantic shift (OOD: Novel classes).

## Appendix F. Examples of Pixel-Space Outliers

Figure 5 provides examples of synthetic outliers generated from the ISIC2019 and NCT
training data respectively. The first four rows denote examples of `Augmix o Jigsaw` while
the remaining rows provide examples of `RandConv` with large kernel sizes.

## Appendix G. Fine-grained results for MedMNIST

Figure 4 summarizes the performance of different calibration strategies in terms of balanced
accuracy (average of sensitivity and specificity) and AUROC scores for modality shift and
novel class detection. Further, in Table 4, we provide the AUPRIn scores for novel class
detection for each of the MedMNIST datasets. In general, we find that our approach
significantly outperforms the baselines except in the case of PathMNIST and DermaMNIST.

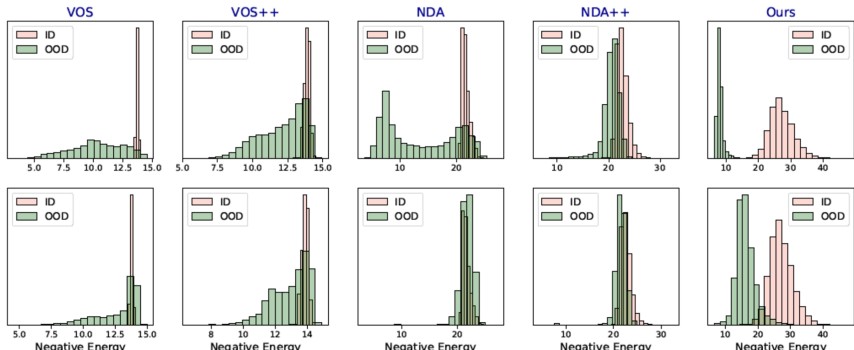

Figure 7: **Histograms of negative energy scores.** With OrganaMNIST as ID, the top row corresponds to modality shift (OOD: TissueMNIST) and the bottom row shows semantic shift (OOD: Novel classes).

Tables 5 - 12 provide the AUROC/AUPRIn scores for modality shift detection obtained with each of the MedMNIST datasets.

## Appendix H. Fine-grained results for ISIC2019 and Colorectal Cancer Benchmarks

In Tables 13 and 14, we provide the AUROC scores for modality and shift detection on the ISIC and the Colrectal Cancer dataset against the OOD benchmarks described earlier.

## Appendix I. Additional Histograms of Negative Energy Scores

Figures 6 and 7 (first row) depict the histograms of the negative energy scores where DermaMNIST and OrganaMNIST are used as ID, and OrganaMNIST nd TissueMNIST are used as modality shifts respectively. The second row corresponds to the histograms associated with the novel class detection in each case. We find that our approach produces well-separated distributions and much higher scores for ID data in all examples.

