# OpenReview forum: "Know Your Space: Inlier and Outlier Construction for Calibrating Medical OOD Detectors"
_MIDL.io/2023/Conference — MIDL 2023 Oral_

### Official Review · Reviewer_ZZtJ · 2023-02-01

**Confidence:** 4
**Preliminary Rating:** 3
**Recommendation:** Poster

**Summary:**

The paper proposes to improve/ better calibrate out-of-distribution detection with the inclusion of synthesized pixel-space and latent-space inliers and pixel-space and latent-space outlier. They compare their methods against methods which do some of the inlier/outlier synthesis methods and to ODIN in the "novel class" and "modality shift" setting show better performance in most cases.

**Strengths:**

+ The idea to use pixel-space inliers and pixel-space outliers and latent-space in/and outliers at the same time seems like a good idea (while not novel individually).
+ The experiment setup with the number of datasets and types of shift is quite extensive and the presented figures are quite nice this amount of experiments could/should be adapted as the standard in such a setting.

**Weaknesses:**

- While the authors combine the different synthesized samples for better OOD performance, I feel like there is not that much methodological novelty.
- The authors state the hyperparameters ($m_{ID}$, $m_{OOD}$, $\alpha$, $\beta$) , without saying how they chose these values. This always makes me a little suspicious where some implicit hyperparameter fine-tuning on the test sets was done (by accident). However, in such a OOD setting it is particularly important (since as the authors also state), we do not want to have any bias towards certain anomalies in the results.
- I feel like a ablation study is missing. Given the experiment I find it really hard to say what the main benefit is and what causes the good performance (and what the main contribution of the paper is). For NDA, the performance seems to drop when including the additional pixel-space inliers ? Perhaps the same happens for your method. So while the comparison to NDA and NDA++ and VOS and VOS++, I feel like a analysis/ablation study of the contribution of each combination pixel&latent space inliers&outliers would greatly strengthen the paper and also make the contribution of this paper way clearer ( i.e. normal model (NM), NM + pix in, NM + pix in + pix out, NM + pix in + lat in, NM + pix in + lat out, NM + pix in + lat in + lat out , ... )

**Deanonymize Review:**

no

**Detailed Comments:**

Perhaps in Fig 2 make the range fixed and also have the method names on each plot...

**Paper Type:**

methodological development

**Questions To Address In The Rebuttal:**

The authors should state how they came up with their hyperparamters and if any data set was used to determine the hyperparamters, which data set was used.
Additionally it would be really great include the ablation study, otherwise I find it hard to judge the main contribution of this paper (since the methodological novelty is not that great, the analysis should be a bit thorough). Adding this would/could greatly change my opinion of this paper (as I said I liked the number of experiments and the idea).

---

### Official Review · Reviewer_mCQZ · 2023-02-02

**Confidence:** 3
**Preliminary Rating:** 4

**Summary:**

The authors propose a method for performing OOD detection using a classifier trained on synthetic outliers. The authors suggest it most effective to train on inliers synthesised in latent space and outliers synthesised in pixel space, and test this by comparing to a number of existing methods on a range of datasets.

UPDATE: i'm happy with the author reponses and am leaving my rating as weak accept.

**Strengths:**

- The authors evaluate on datasets at both low and higher resolutions - a strength as many methods in the computer vision literature limit themselves to only evaluating on 32x32 images.

- The authors evaluate on some challenging near-OOD problems

- The method outperforms competing performs on a range of benchmarks

- The paper is clearly written

**Weaknesses:**

I think the background failures to fully contextualise this family of methods within the OOD detection literature. The introduction states that it is necessary to calibrate detectors using some form of OOD data. However, there are plenty of methods that seek to classify OOD samples while only trained on in-distribution data. There are a number of likelihood-based methods, of which the currently best performing is probably [1]. There are also methods that seek to develop unsupervised representations of the data using contrastive learning and detect OOD samples with these representations [2,3].

Related to the previous point, I'm not convinced that G-ODIN is the current state-of-the-art in OOD detection without outlier exposure. CSI [2] reports good performance on some very challenging benchmarks, such as CIFAR10 vs CIFAR100 - though unfortunately doesn't directly compare against G-ODIN. I think it would strengthen the work to benchmark against [2].

[1] Density of states estimation for out of distribution detection, Morningstar et al

[2] SSD: A UNIFIED FRAMEWORK FOR SELFSUPERVISED OUTLIER DETECTION

[3] CSI: Novelty Detection via Contrastive Learning
on Distributionally Shifted Instances

**Deanonymize Review:**

no

**Detailed Comments:**

The authors state "this paper makes a striking finding that the space in which the inliers and outliers are synthesized". I found it difficult to fully evaluate this claim, as it wasn't clear to me if the other methods tested (VOS, NDA and variants thereof) hold absolutely everything else held constant with respect to the proposed method apart from the space that the outliers are synthesised in, or if these methods contain other differences too.

Is the balanced accuracy a measure of the classifier performance: that is, the ability to assign in-distribution samples to their correct classes? It isn't clear to me if it is this or another measure of the labelling of images as in-distribution vs OOD. If the former, perhaps it could be clarified in the text; if the latter than this paper should include some measures of in-distribution classifier performance.

I didn't see the image size of the ISIC2019 dataset mentioned in the experimental details. As stated above I think it's a strength the authors evaluate on a range of image sizes so this information would be welcome.

A couple of matters of personal taste:
- I found the presentation of results in Figure 2 made it difficult to compare methods and would much prefer a table as the primary way of presenting them, perhaps with Figure 2 in the supplementary.

- The use of red colouring to highlight second best results in Figure 1 is jarring - I, and I imagine others, might imagine that green represents best and red worst.

**Paper Type:**

methodological development

**Questions To Address In The Rebuttal:**

Can the authors provide evidence that G-ODIN is the SOTA in OOD detection without exposure? If not I think some of the papers I've mentioned should be included as baselines.

I'd like to be sure that the experiments really do support the claim "this paper makes a striking finding that the space in which the inliers and outliers are synthesized".

---

### Official Review · Reviewer_npsu · 2023-02-05

**Confidence:** 5
**Preliminary Rating:** 5
**Recommendation:** Best Paper Award, Oral

**Summary:**

Authors propose a particular scheme of synthetic inliner and outlier sample generation for the energy-based OOD detection approach. Highlighting the fact that synthetic sample generation is a key approach for methods that rely on exposure, authors study what type of generation makes the most sense. Experiments with a diverse set of image sets show that the resulting model is very strong.

**Strengths:**

- Excellent analysis leads to a strong and sound method.
- Amazing results. Truly impressive.
- Extremely well written article.
- I am not sure what I have to write more. This article is REALLY good in my opinion.

**Weaknesses:**

- The method may seem like not the most novel method, however, I believe this is far from the truth. Authors' analysis is spot on. Their findings and how they use it is excellent.
- Once again, not sure what to write to fill the minimum character count. I think this is a really good article.

**Deanonymize Review:**

no

**Paper Type:**

methodological development

**Questions To Address In The Rebuttal:**

- It would be very nice to show some visual results towards what type of images the latent space and pixel-space methods yield.
- It would be nice to discuss a bit further the link with methods based on Mahalonobis distance.

---

### Meta-Review · Area_Chair_jfno · 2023-02-25

**Recommendation:** Accept (Oral)
**Confidence:** 5

**Metareview:**

The paper provides an interesting study of  out-of-distribution detection by including synthetic inliers and outliers. There is consensus among the reviewers that the paper is worth publication at MIDL. The analysis is convincing, the experiments are quite comprehensive and the results are quite strong. Also, the paper is very well-written and the authors provided detailed responses during the discussion. Therefore, I recommend acceptance.